# Induced photoelectron circular dichroism onto an achiral chromophore

Etienne Rouquet[1,2], Madhusree Roy Chowdhury[1], Gustavo A. Garcia [1],
Laurent Nahon [1] ✉, Jennifer Dupont[2], Valéria Lepère [2],
Katia Le Barbu-Debus [2] & Anne Zehnacker [2] ✉

An achiral chromophore can acquire a chiral spectroscopic signature when interacting with a chiral environment. This so-called induced chirality is documented in electronic or vibrational circular dichroism, which arises from the coupling between electric and magnetic transition dipoles. Here, we demonstrate that a chiroptical response is also induced within the electric dipole approximation by observing the asymmetric scattering of a photo-electron ejected from an achiral chromophore in interaction with a chiral host. In a phenol–methyloxirane complex, removing an electron from an achiral aromatic π orbital localised on the phenol moiety results in an intense and opposite photoelectron circular dichroism (PECD) for the two enantiomeric complexes with *(R)* and *(S)* methyloxirane, evidencing the long-range effect (~5 Å) of the scattering chiral potential. This induced chirality has important structural and analytical implications, discussed here in the context of growing interest in laser-based PECD, for in situ, real time enantiomer determination.

Photoelectron circular dichroism (PECD) is a chiroptical spectroscopic technique, i.e. a spectroscopy resting on the interaction between circularly polarised light (CPL) and a chiral molecule. Defined as a forward-backward asymmetry in the photoelectron angular distribution after ionisation of a chiral system by a CPL, it depends on both the initial state, i.e. the orbital from which the electron is ejected, and the final state, i.e. the electron continuum and therefore on the photoelectron kinetic energy, and the scattering chiral potential of the cation. As such, it is very sensitive to structural or conformational isomerism[1-3] and to complex formation[4,5].

The normalised one-photon photoionisation angular distribution function $I^{\{p\}}(\theta)$ is written as:

$$I^{\{p\}}(\theta) = b_0(1 + b_1^{\{p\}}P_1(\cos\theta) + b_2^{\{p\}}P_2(\cos\theta)) \quad (1)$$

$P_1$ and $P_2$ are the first and second Legendre polynomials, $p$ defines the ionising light polarisation, with $p = 0$, $+1$, and $-1$ for linear, left- and right-handed circular polarisation, respectively; $\theta$ is the angle of electron emission relative to the circularly polarised light propagation axis. The $b_0$ coefficient is the (isotropic) photoelectron spectrum (PES), the

$b_2^{\{\pm 1\}}$ coefficient is $-1/2b_2^{\{0\}}$, $b_2^{\{0\}}$ being the usual (achiral) anisotropy parameter in photoionisation with linearly polarised light, commonly known as β, which is not studied here. The dichroic $b_1$ coefficient vanishes for linear polarisation or non-chiral molecules. For chiral molecules, $b_1^{\{1\}} = -b_1^{\{-1\}}$, so that opposite effects are observed for right- and left-handed CPL. Opposite effects are also observed for opposite enantiomers of a chiral molecule, as expected for any chiroptical technique.

PECD has been measured for both valence and core electrons of neutral molecules, as well as for anionic species[6-11]. Very large dichroic coefficients have been measured even for core electrons or electrons extracted from localised valence orbitals remote from the chiral centre, which unambiguously shows the importance of rather long-range effects in PECD[9,12,13]. Other chiroptical methods also exhibit long-range effects, such as electronic and vibrational circular dichroism, for which induced chirality has been evidenced[14-17]. It has been postulated as well in the case of Raman Optical Activity in an achiral solvent[18]. This term refers to the appearance of a circular dichroism signal in an achiral chromophore. Although it has been well documented for electronic circular dichroism (ECD) in the gas phase and for host-guest systems,

[1]Synchrotron SOLEIL, L'Orme des Merisiers, Départementale 128, F-91190 St Aubin, France. [2]Institut des Sciences Moléculaires d'Orsay (ISMO), CNRS, Université Paris-Saclay, F–91405 Orsay, France. ✉e-mail: laurent.nahon@synchrotron-soleil.fr; anne.zehnacker-rentien@universite-paris-saclay.fr

biomolecules, perovskite nanostructures, etc., the theoretical description of the non-local effects that couple the host and the environment is still challenging[19–23]. Non-local effects are also important in vibrational circular dichroism (VCD)[15], in particular in the solid state where they were studied recently from a theoretical point of view[24,25]. In VCD, as it is also the case for ECD, the intensity and sign of the signal is given by the scalar product of the electric and magnetic transition dipole moments[26]. A significant non local effect is direct coupling, which is the coupling between the electric moment located on one sub-unit and the magnetic moment of another molecular sub-unit[24]. This coupling may explain induced chirality that has been observed in VCD, for example a chloroform solution containing camphor[27].

In contrast, PECD stands out compared to the other chiroptical methods by the fact that it is allowed in the pure electric dipole approximation. It is therefore much more intense, up to ~40%[28], and as such very-well adapted to dilute media and in particular to the direct observation of chiroptical effect in weakly-bound complexes at the molecular level. Several experimental and theoretical attempts have addressed questions regarding the range of PECD, with strong effects measured in achiral orbitals not located at the chiral centre[7,9,13,29,30]. These efforts, however, have been limited to intramolecular distances, for both valence shell and inner shell PECD. Here, we focus on the complexes formed between phenol (Phe), a non-chiral chromophore, and methyloxirane (MOx), whose structures are shown in Fig. 1a, b. We aim to assess the range of the influence of the chiral host on PECD across intermolecular distances, leading to an induced PECD on the achiral moiety.

MOx is a rigid chiral host, whose ionisation energy (IE) and PECD are known from previous studies[29,31,32]. The choice of the systems is dictated by the fact that the two molecules have significantly different ionisation energies (IEs). Phenol has a relatively low ionisation energy compared to methyloxirane (8.49 vs. 10.25 eV). Therefore, the orbitals

of the chromophore and those of the ligands are well separated and it is possible to selectively ionise those orbitals localised on the achiral moiety. Moreover, the possible OH…O hydrogen bond from the phenol to the MOx ensures easy formation of complexes and strong interaction between the chiral and achiral parts. The structure of the complex is determined by means of double resonance IR-UV spectroscopy combined to quantum chemistry calculations. We show PECD signals obtained at photon energies of 8.5 and 10.4 eV for the electrons recorded in coincidence with the ions at the mass of the Phe:MOx complex, resulting from the ionisation of phenol orbitals within the complex. This effect appears as induced chirality, since the achiral chromophore appears "chiral" due to the influence of the chiral host in the complex.

## Results

### Structure of the complex

The REMPI $S_0$-$S_1$ spectrum of the complex is shown in Fig. 1c and is reminiscent of that observed for the Phe:oxirane complex[33]. The origin of the $S_0$-$S_1$ spectrum (35960 cm$^{-1}$) is shifted by -404 cm$^{-1}$ relative to the bare phenol $S_0$-$S_1$ origin, a shift very similar to that of the complex between phenol and the related achiral compound oxirane (-358 cm$^{-1}$) that was due to one conformer only[33]. The most intense band is followed by a progression built on a low-frequency mode (26 cm$^{-1}$), similar to that observed at 26.9 cm$^{-1}$ in Phe:oxirane and assigned to the hydrogen bond bending motion. An intense band appears at 165 cm$^{-1}$ of the origin, which is similar to that at 188 cm$^{-1}$ in Phe:oxirane, assigned to the hydrogen bond stretch.

These results suggest that the Phe:MOx complex exists as a single conformer. This hypothesis is confirmed by the IR-UV double resonance spectra recorded with the probe set on the most intense bands of the spectra, namely, the origin, the bands at 26 and 165 cm$^{-1}$. They are shown in Fig. 1d, together with that obtained by probing the origin

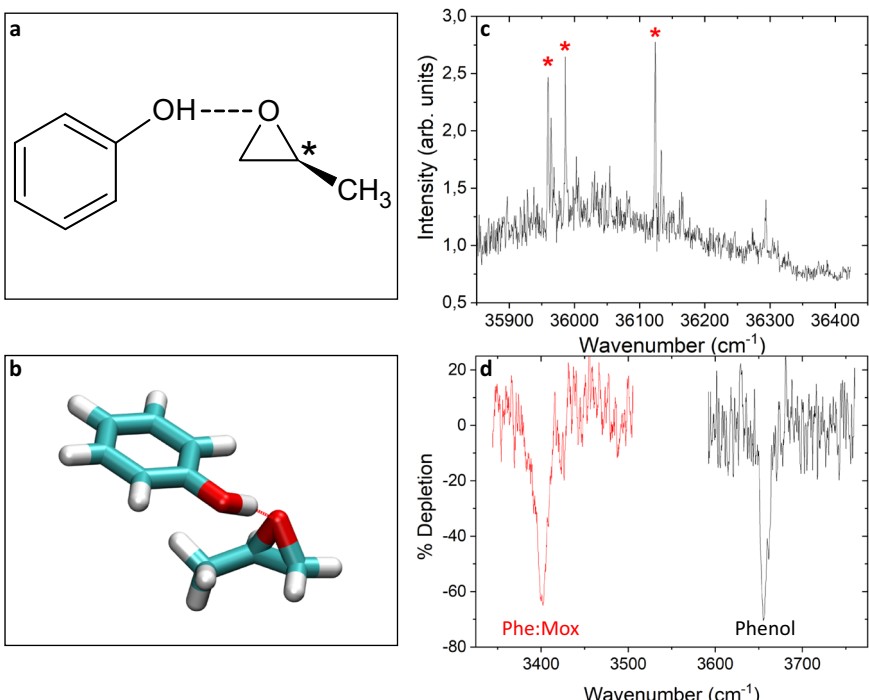

**Fig. 1 | Structure of the phenol (Phe) (S) methyloxirane (MOx) complex.**
**a** Schematic view of the complex. The stereogenic centre is indicated by an asterisk.
**b** Most stable structure of the Phe-MOx complex calculated at the B3LYP-D3BJ/6-311 + + G(d,p) level of theory. The hydrogen bond is indicated by a dotted line.
**c** Electronic spectrum of the Phe-MOx complex recorded by using the resonance-enhanced multi photon ionisation technique. The bands marked with asterisk (*)

were probed in IR-UV experiments. **d** Vibrational spectrum of isolated phenol and of the Phe-MOx complex recorded with the IR-UV double resonance technique by setting the probe on the transition origin of bare phenol and of the Phe-MOx complex, respectively. Source data are provided as a Source data file.

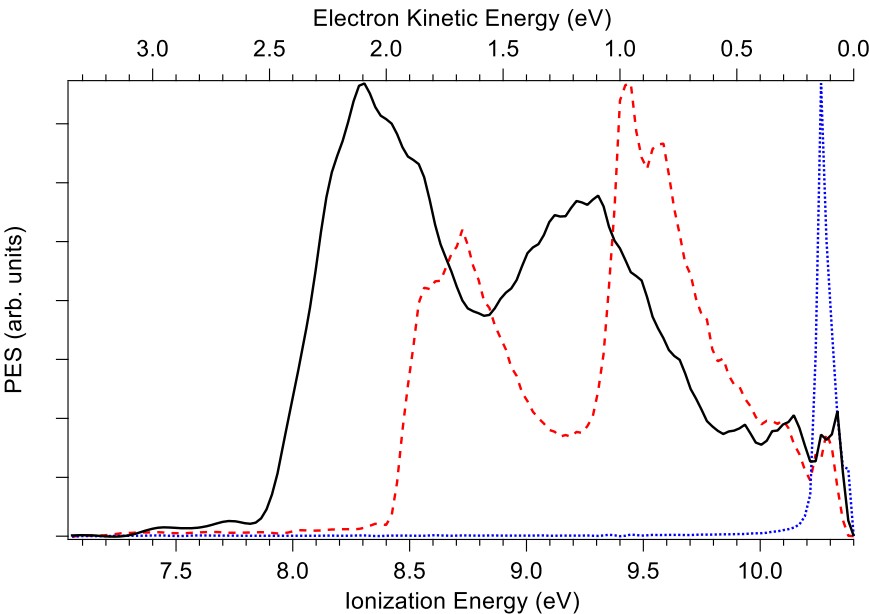

**Fig. 2 | Photoelectron spectra.** Comparison between the photoelectron spectrum of bare phenol (red dashed line), MOx (blue dotted line) and the Phe:MOx complex (black line), recorded at a photon energy of 10.4 eV, slightly above the adiabatic ionisation energy of MOx (10.24 eV)[29]. Source data are provided as a Source data file.

transition of bare phenol. Probing either the complex origin transition or the band at 26 or 165 cm⁻¹ results in the same IR spectrum. It shows a single ν(OH) transition located at 3401 cm⁻¹, shifted down in energy by 256 cm⁻¹ relative to the free ν(OH) of phenol (3657 cm⁻¹)[34] as expected for a hydrogen bonded system. This value is reminiscent of the ν(OH) frequency measured in other complexes involving phenol in interaction with an ether[35,36].

The most stable complex is very similar to the Phe:oxirane complex and is shown in Fig. 1b. Like all the structures of the complex calculated below 1 kcal/mol, shown in Supplementary Fig. 1, it involves a hydrogen bond from the phenol hydroxyl to the MOx oxygen (OH···O distance of 1.82 Å). Besides stability considerations, it is also the calculated complex with the ν(OH) the closest to the experimental value (3396 cm⁻¹) and we therefore assign the observed complex to this structure. Based on the calculated frequencies, the experimental bands at 26 and 165 cm⁻¹ are assigned to the hydrogen bond rocking and stretching motions, calculated at 27 and 167 cm⁻¹, respectively.

The two oxygen lone pairs are equivalent in oxirane but are not in MOx, due to the methyl group that makes the molecule chiral. The oxygen atom can therefore be seen like a stereogenic centre due to complexation with phenol. However, the complex involving a hydrogen bond with the other doublet is higher in energy by 0.28 kcal/mol and its calculated frequency (3387 cm⁻¹) does not match that observed here (see Supplementary Fig. 1). The preference for one of the two prochiral lone pairs probably results from additional dispersion interaction between the aromatic ring and the methyl that stabilises the structure to which the experimentally observed complex is assigned. However, we cannot rule out the hypothesis that bands of lesser intensity, which have not been probed in IR-UV experiments, are due to the other conformer.

## Photoelectron spectrum and nature of the orbitals

The photoelectron spectrum (PES), recorded by monitoring the electrons in coincidence with the ions at the mass of the Phe:MOx complex (m/z 152) and at that of phenol (m/z 94), is shown in Fig. 2, using a photon energy of 10.4 eV.

The observed adiabatic ionisation energy (AIE) of the Phe:MOx complex is 7.8 eV, of the same order as that measured by zero electron kinetic energy (ZEKE) spectroscopy for the similar system

Phe:dimethylether complex (7.76 eV), in line with the similar hydrogen bond from the phenol hydroxyl to the ether oxygen[37].

The two bands observed in the PES of the complex at vertical ionisation energies (VIE) of 8.3 and 9.2 eV correspond to the ionisation of the HOMO and HOMO-1, respectively, which are π orbitals localised on the aromatic ring (vide infra). These values are obtained by fitting the experimental photoelectron spectra by one or two Gaussian functions, depending on the photon energy[12]. Comparison to the PES of pure phenol shows that the VIE of the complex is shifted down by -0.5 eV relative to that of phenol. The calculated structures, one-electron density plots and VIEs for the three outermost orbitals are shown in Fig. 3. The calculated values of 8.11 and 8.93 eV are in good agreement with the abovementioned experimental values as is the calculated downshift of 0.42 eV in the VIEs of the HOMO phenol orbital upon complexation. Most importantly, we observe that for the HOMO and HOMO-1 orbitals, the electron wave function of the complex is achiral and localised on the phenol moiety, the MOx moiety being a mere spectator. The electron density has been plotted with an isovalue of 0.01, i.e. much lower than the typical 0.07 value used here, and is shown in Supplementary Fig. 2. The plot shows a tiny contribution of the σ orbital of the methyloxirane CH interacting with the phenol aromatic ring, which can be neglected.

The HOMO-2 orbital of the complex is the lone pair located on the MOx oxygen atom. It is calculated at 11.48 eV in the complex, vs. 10.72 eV in bare MOx. The upshift in energy upon complexation is due to the hydrogen bond formation that destabilises the oxygen lone pair[31]. None of the photon energies chosen for the experiments described here allow ionising the MOx moiety orbitals within the complex, so that all the electrons considered in this study originate from the achiral orbitals localised on the phenol moiety of the complex (vide infra Fig. 4).

## PECD results

The PECD was measured for photon energies ranging from 8.5 eV to 10.4 eV. Before doing so, the conditions were carefully adjusted to optimise the signal of the complex in the mass spectrum (see Supplementary Fig. 3). The mirroring between the enantiomers was checked at several photon energies, as well as the absence of PECD on bare phenol. We shall discuss here PECD obtained for selected photon

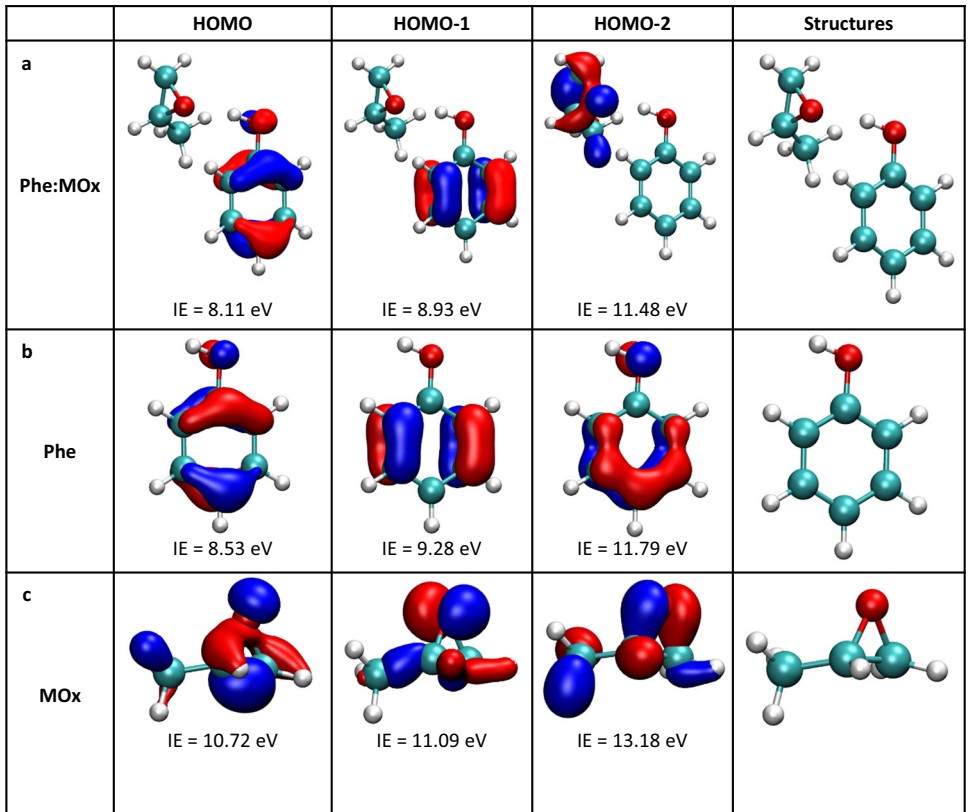

**Fig. 3 | Localisation of the frontier orbitals Frontiers orbitals starting from the highest occupied molecular orbital (HOMO) calculated for the Phe:MOx complex (a) as well as bare phenol (b) and bare methyloxirane (c) at the MP2/6-31++G(d,p) level.** The electronic density was plotted with an isodensity value of 0.07. Calculated vertical ionisation energies (IE) are obtained with the outer valence Green's function (OVGF) method and cc-pVTZ basis set. Source data are provided as a Source data file.

energies, namely, 8.5 and 10.4 eV. The results obtained at 9.0 and 9.7 eV are shown in Supplementary Figs. 4 and 5 of the Supplementary Information.

Figure 4a shows the results obtained at a photon energy of 10.4 eV and filtered at the mass of the complex at *m/z* 152. The PES shows the bands at 8.3 and 9.2 eV (i.e. electron kinetic energies of 2.1 and 1.2 eV) corresponding to the ionisation of the phenol π orbitals, as already mentioned. The weaker signal above the 10 eV binding energy corresponds to the onset of the HOMO-2 of the complex, which is the lone pair located on the MOx oxygen atom. The obtained PECD shows, within our error bars, quasi-perfect mirroring between the values measured for the complexes of phenol with the *(R)* and *(S)* enantiomers of MOx. The PECD reaches an almost constant value of ~3 % and keeps the same sign all over the band corresponding to the HOMO localised on the phenol moiety. It then changes sign with a clear extremum at 8.8 eV. The PECD of bare phenol, which can be ionised at these photon energies, is zero as expected signalling negligible systematic errors.

The most telling result is that obtained at a photon energy of 8.5 eV, i.e. below the ionisation energy of isolated phenol, shown in Fig. 4c. A clear PECD-asymmetry is evidenced at this ionisation energy, with the expected specular relation between the complexes of phenol with the *(R)* and *(S)* enantiomers of MOx. The mean PECD measured for the first band of the PES (electron kinetic energy in the 0−0.5 eV range) is of the same sign as that obtained at 10.4 eV, but slightly weaker, of the order of 2%. Note that the PECD is not perfectly flat across the electronic band. This smooth variation is attributed to vibronic transitions leading to a production of photoelectrons with a broad energy distribution, combined with the dependence of PECD on electron kinetic energy[38].

These measurements unambiguously evidence induced PECD during photoionisation from the HOMO orbital of the hydrogen-bonded phenol. As mentioned above, the orbital itself is fully localised on phenol, and is not chiral. The origin of the induced PECD observed should stem from the chiral potential that is experienced by the scattered photoelectron, in the same manner as this drives the observed large PECD-induced asymmetries from purely spherical and therefore achiral core orbitals of chiral species[9,13]. Remarkably, the range of this scattering effect is here well beyond the typical ~1 Å distance between such achiral orbital and the chiral centre, with the distance between the centre of the aromatic ring of the phenol moiety and the stereogenic centre, the asymmetric carbon atom of MOx, being ~5 Å. A recent report of photoelectron elliptical dichroism of chiral aromatic alcohols with increasing distances between the benzene moiety from which the photoelectron departs and the chiral centre indicates that the asymmetry parameter decreases when this distance increases[30]. Another major difference, as compared to core-PECD within free chiral molecules, lies in the fact that we are dealing here with a complex with two clearly distinct moieties. Therefore, the induced PECD observed on the phenol achiral moiety appears as a manifestation of induced chirality. Indeed, complexation with the chiral MOx host offers the phenol departing electron a chiral scattering potential.

We can also compare the PECD of the HOMO and the HOMO-1 orbitals as a function of the electron kinetic energy, as shown in Fig. 4 d. While the PECD of both orbitals has the same sign at low kinetic energy, they are of opposite signs at higher electron kinetic energy, with the HOMO showing a much larger PECD magnitude.

The Phe$_2$:MOx complex containing two phenol units and one chiral MOx molecule also appears in the mass spectrum recorded at a

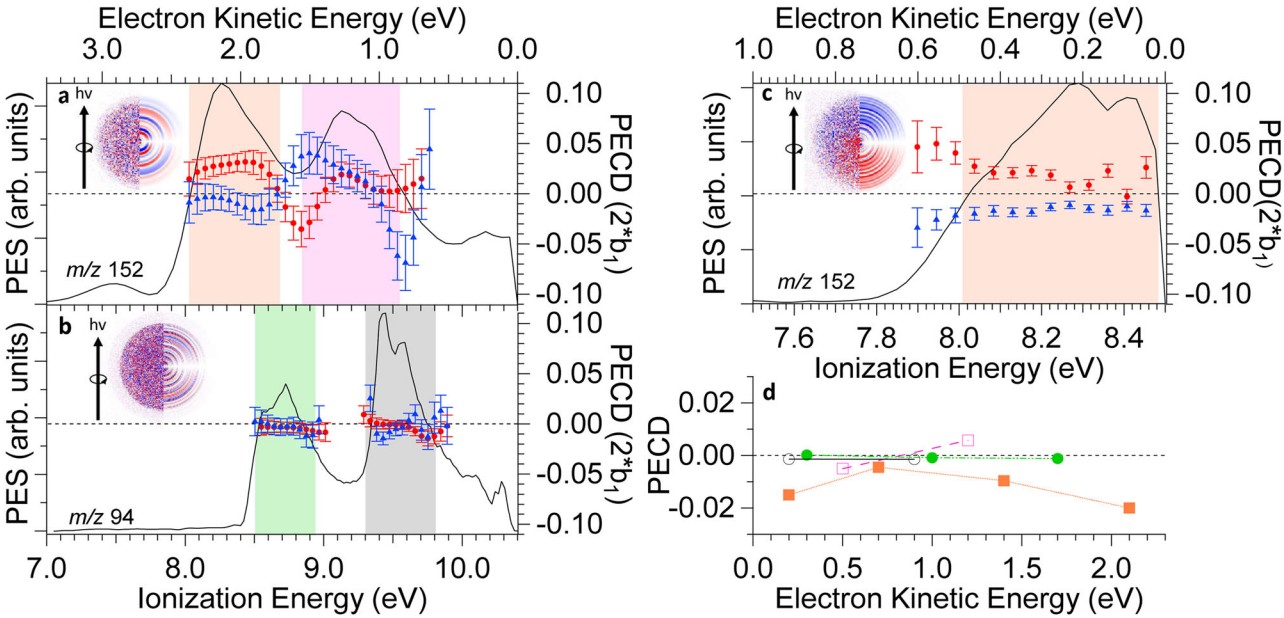

**Fig. 4 | PES and dichroic parameter at 10.4 eV and 8.5 eV for the complex of phenol with the two enantiomers of methyloxirane. a** The photoelectron spectra (PES) (solid line) and photoelectron circular dichroism (PECD) for the complex of phenol with (S) MOx (red circles) and (R) MOx (blue triangles) are recorded in coincidence with the ions at *m/z* 152 at a photon energy of 10.4 eV. The zones corresponding to the HOMO and HOMO−1 of the complex are highlighted in orange and pink, respectively. The insert shows raw (left part) and Abel-inverted (right part) difference between left (LCP) and right (RCP) circularly polarised light (LCP−RCP) images, corresponding respectively to the 2D projection and an equatorial slice of the 3D velocity distribution filtered at the mass of the Phe:(R)MOx complex, obtained with the velocity map imaging (VMI) spectrometer. The direction of the light is vertical, from the bottom to the top of the image and is indicated by an arrow. **b** Same for bare phenol, recorded in coincidence with the ions at *m/z*

94. The zones corresponding to the HOMO and HOMO−1 of bare phenol are highlighted in green and grey, respectively. As expected for an isolated achiral chromophore, the value of the PECD is zero. **c** Same for the complex at a photon energy of 8.5 eV. Neither bare phenol nor MOx can be ionised at this energy value. **d** Mean values of the PECD weighted by the PES intensity of the two highest occupied orbitals HOMO and HOMO−1 of the Phe:(R) MOx complex as a function of the electron kinetic energy recorded at the photon energies of 8.5, 9.0, 9.7 and 10.4 eV. The shown values are half the difference between those obtained for the Phe:(R)MOx and Phe:(S)MOx complexes. The colour code is the same as that used in **a**–**c**, namely, green full circles and grey empty circles for the HOMO and HOMO-1 of phenol, respectively, and orange full squares and pink empty squares for the HOMO and HOMO-1 of the complex, respectively. Error bars correspond to statistical (standard deviation) errors. Source data are provided as a Source data file.

photon energy of 8.5 eV (see Supplementary Fig. 3). The most stable calculated structures of the Phe$_2$:MOx complex are shown in Supplementary Fig. 6 and consist in a phenol dimer hydrogen-bonded to the MOx. Although the detailed study of this cluster goes beyond the scope of this article, it is worth mentioning that PECD is clearly observed for the complex, with the expected mirroring relation between the two enantiomers of MOx, as shown in Supplementary Fig. 7. Note that the overall PECD magnitude does not seem to decrease when the size of the complex increases, analogously to past and recent experiments that have demonstrated that PECD persists upon aggregation, even up to the formation of nanoparticles, due to factors such as the narrowing of the conformer population and the emergence of local ordering[39].

In conclusion, this first example of the manifestation of induced chirality in PECD on an achiral chromophore by complexation with a chiral tag underlines the role of the long-range chiral scattering potential extending to intermolecular distances, here up to ~5 Å. From the point of view of the achiral moiety, the presence of the chiral MOx host offers to the phenol departing electron a chiral scattering potential, leading to induced PECD. This induced PECD provides a sensitive chiroptical probe of the molecular structures, for instance in the case of multiple conformations.

Conversely, from the point of view of the chiral moiety, the induced PECD on the achiral (phenol) chromophore offers innovative analytical opportunities. The study of PECD has been pioneered in the one-photon regime with synchrotron radiation about two decades ago[40]. With the first demonstrations of laser-based multi-photon (MP) PECD in 2012[41,42], the field has been blooming. It now reaches such a

level of reliability that using laser-based MP-PECD as a table-top analytical tool to retrieve in situ enantiomeric excesses in the gas phase has been suggested by different groups[43–48]. However, because of the ionisation scheme used involving near/mid-UV range photons, these studies have been so far limited to terpenes or to chromophore-bearing species such as phenylalanine[49]. As we demonstrated here, the complexation with an achiral chromophore could be a way to extend such laser-based chiroptical measurements to a large array of chiral species which cannot be excited in the near/mid UV, including for instance non-aromatic amino-acids, whose PECD has been so far studied only by one-photon synchrotron ionisation, a technique which unlike laser MP-PECD, does not allow a direct selection of conformers[3,39,50].

The chirality induction process observed in PECD, at the core of the present study, appears universal and does not depend on the chemical bond between the moiety from which the photoelectron is ejected and the moiety containing the chiral centre, i.e. in the case of complexes, between the chromophore and the chiral host. Therefore a similar effect of induced PECD should also be observed on systems involving weaker interactions than hydrogen bonds, for example the benzene:MOx system. The transmission range of chirality is also an open question and longer distances between the chiral and achiral parts of the complex, for example by using chemically-modified rigid scaffolds, are of great interest and their study are planned in a near future. Time-resolved photoelectron circular dichroism experiments would be very useful to further assess the spatial range of induced chirality in a molecular complex. Note that chiral changes upon covalent-bond dissociation of a chiral molecules have been recently

observed by time-resolved PECD[51]. The formation of a new chiral centre arising from complexation of a prochiral molecule is also of interest and will be studied by the resonance-enhanced two-photon ionisation PECD scheme recently developed in our group, which will be reported in a near future.

## Methods

### Experimental methods

Commercially available samples of phenol and enantiopure MOx were purchased from Sigma-Aldrich. Phenol was brought into the gas phase by resistive heating in an oven at 75 °C. MOx was kept at a temperature of −30 °C during the whole experiment.

The UV and conformer-selective IR spectra were obtained at ISMO. The pulsed supersonic beam was produced by expanding 2 bar of helium or neon through a 200 μm pulsed nozzle (General Valve - Parker)[52]. Mass-resolved electronic spectra were obtained using one-colour resonance-enhanced two-photon ionisation (RE2PI). The UV source was a frequency-doubled dye laser (Sirah equipped with C540A dye) pumped by the second harmonic of a Nd:YAG laser (Surelite III - Continuum). It crossed the skimmed supersonic beam (skimmer diameter of 500 μm) in the interaction zone of a linear time-of-flight (TOF) mass spectrometer (Jordan, one-metre length). The ion signal was detected by a microchannel plate detector (RM Jordan, 25 mm diameter), averaged by an oscilloscope (Lecroy wavesurfer), and processed through a personal computer. Vibrational spectra were recorded using the IR-UV double resonance method[53,54]. The slightly focused (0.5 m focal length lens) IR laser beam (OPO/OPA - Laser Vision) was counter-propagated relative to the UV laser beam and superimposed to it in the source region. After fixing the UV probe laser on selected vibronic transitions of the electronic spectrum, the wavelength of the IR pump laser was scanned in the 3 μm region. The IR absorption was then detected as a depletion of the UV-induced ion signal. The IR pulse was triggered ~80 ns before the UV pulse. Synchronisation between the lasers was controlled by a homemade gate generator. The IR spectra were recorded resorting to an active baseline scheme, by measuring the difference in ion signal produced by successive UV laser pulses (one without and one with the IR laser pulse present)[55].

The PECD experiments were performed at the beamline DESIRS of the synchrotron facility SOLEIL. This undulator-based beamline allows obtaining tuneable polarisation-controlled vacuum ultraviolet (VUV) radiation, with an absolute circular polarisation rate above 97%[56]. It is equipped with the dedicated SAPHIRS set-up including a continuous supersonic expansion and the i²PEPICO DELICIOUS3 double imaging spectrometer, which allows detection of electrons and ions in coincidence[57].

Phenol was expanded through a 200 μm nozzle, using 0.5 bar of helium as a carrier gas seeded with MOx. The ions and electrons formed at the intersection of the VUV photon beam with the doubly skimmed molecular beam were accelerated in opposite directions and detected in coincidence by a modified Wiley - McLaren imaging time of flight (TOF) spectrometer and a velocity map imaging (VMI) spectrometer, respectively.

Between 20 and 50 successive files accumulated for 10 min each were recorded with alternating light helicity for the complexes obtained with each enantiomer of MOx. The corresponding raw electron images were filtered at the mass of the complex ($m/z$ 152) and processed via the p-Basex software[58]. The p-Basex software is based on the Abel transform of the electron distribution and aims to reconstruct the three-dimensional Newton sphere of the expanding electrons from its two-dimensional projection. It consists of fitting a set of polar basis functions with a known and exact inverse Abel function. The sum (LCP + RCP) image provided the PES and the difference (LCP-RCP) image provided the $b_1$ parameter and therefore the PECD= $2b_1$[59]. The error bars were estimated assuming a Poisson distribution of the image pixel intensities and propagating the Poisson standard deviation

through the image inversion transformation. This is done by assuming that each individual pixel of the image has a variance of $N$, $N$ being the number of counts. All the operations leading to the PECD propagate this variance. For instance, subtracting the background increases the variance by $B$ ($B$ being the number of counts of the background). The algebra operations of the Abel transform also propagate the variance. The error bars are given as the square root of the final variance.

### Theoretical methods

The potential energy surface (PES) was explored using the OPLS-2005 force fields[60] with the advanced conformational search implemented in the MacroModel suite of the Schrödinger package[61]. The geometry of the most stable structures within 21 kJ/mol were locally optimised using the B3LYP functional and the 6-311 + + G(d,p) basis set[62], including D3BJ dispersion corrections[63,64]. The harmonic frequencies were obtained at the same level of theory and scaled by 0.953 to correct for anharmonicity and basis set incompleteness[65,66]. The scaling factor was defined to reproduce the ν(OH) frequency of phenol. The relative stability of the different isomers of the complex was assessed by their enthalpy ΔH relative to the most stable structure. The structures corresponding to the experiment were re-optimised at the MP2/6-31 + + G(d,p) level. The electronic densities were calculated at the MP2 level and the cube files were generated using the cubegen facility implemented in the Gaussian software, version G16 B01[67]. The electronic density of the frontier orbitals was plotted for each occupied orbital considered using the VMD software, with an isodensity value of 0.07[68]. Energies of the valence orbitals were calculated using the outer valence Green's function (OVGF) method and cc-pVTZ basis set, at the MP2/6-31 G + +(d,p) optimised geometries[69,70].

## Data availability

The authors declare that the data supporting the findings of this study are available within the paper and its Supplementary Information files. The raw data that support the findings of this study are available from the corresponding authors upon request. Source data are provided with this paper.

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

## Acknowledgements

This work has been supported by the programme "Investissements d'Avenir LabEx PALM" (ANR-10-LABX-0039-PALM) (A.Z. and L.N.). The authors are grateful to the SOLEIL general staff for providing synchrotron beam under proposal No. 20210155 (A.Z.) and 99220172 (L.N.). We acknowledge the computing centre MésoLUM managed by ISMO (UMR8214) and LPGP (UMR8578), University Paris-Saclay (France). Part of the calculations were performed using Cloud resources from the "Mésocentre" computing centre of CentraleSupélec, École Normale Supérieure Paris-Saclay and Université Paris-Saclay supported by CNRS and Région Île-de-France (https://mesocentre.universite-paris-scalay.fr/).

## Author contributions

Conceptualisation of the project: A.Z., L.N. Methodology: G.A.G. and L.N. Experimental investigation: E.R., M.R.C., G.A.G., L.N., J.D., V.L., and A.Z. Data treatment: E.R. and J.D. Visualisation: E.R., J.D., and V.L. Theoretical modelling: K.L.B.-D., E.R., and A.Z. Funding acquisition: L.N. and A.Z. Supervision: L.N. and A.Z. Writing—original draft: A.Z. with input from L.N. and G.A.G. Writing—reviewing and editing: all authors.

## Competing interests

The authors declare no competing interests.
