## [Peer Review File · Nature Communications]

REVIEWER COMMENTS

Reviewer #1 (Remarks to the Author):

The authors describe a carefully planned and prudently performed experiment in which a photoelectron is triggered from an achiral chromophore close to a chiral molecule. To do this, they first use two-photon spectroscopy and double resonance spectroscopy to determine the structure of the molecular complex before using model calculations and photoelectron spectra to find a suitable ionization window that is uniquely attributable to ionization from the achiral chromophore. They observe a PECD that changes sign with the change of enantiomer. In addition, with reference to the supplement, it is reported that the effect can also be clearly observed at a complex with two achiral molecules. The explanation given is scattering from the chiral core with the additional intriguing aspect that the electron starts far from outside the chiral portion of the complex.

The only difficulty I felt while reading was the terminology: from the clearly explained physics, for my understanding

"Induced Photoelectron Circular Dichroism in an Achiral Chromophore" is misleading, and it should instead be explained as "Induced Photoelectron Circular Dichroism triggered by an Achiral Chromophore" (or something in this spirit) to clarify that the PECD effect still comes from the chiral part of the complex. I ask the authors to seriously reconsider this suggestion and make the appropriate changes in the text prior to publication (of course, publication is recommended).

Two further points:

The anion PECD was first observed by Weitzel et al. and should probably be cited instead or in addition to the much later work cited here (P. Krueger and K.-M. Weitzel, *Angew. Chem., Int. Ed.*, 2021, 60, 17861–1786)

In Figure four d, I could not find the color description of the data points either in the caption or in the text.

Reviewer #2 (Remarks to the Author):

The manuscript by Rouquet et al. presents an experimental investigation of photoelectron electron circular dichroism (PECD)

on Phe:(chiral)MOx complexes. The photoionization occurs from the two lowest molecular orbitals (MO) localized primarily

on the achiral phenol moiety. Surprisingly, despite the achirality of these initial MOs, the PECD exhibits a significant

3% amplitude, depending on the enantiomer. This finding suggests that the photoelectron asymmetry arises from the long-range region of the chiral molecular potential- in the 5 angstroms range, experienced during electron scattering before its emission.

In general, I highly value the experimental findings presented by the authors, which unquestionably surpass the novelty threshold of Nat. Com. The paper demonstrates a well-structured organization and provides a compelling discussion on the prevalence of a singular conformer in the jet. The manuscript itself is easily comprehensible, presenting a clear and coherent narrative throughout. In my opinion, this work unquestionably merits publication and holds immense potential for opening up intriguing avenues in the field of chiro-optical identification of complexes.

List of specific comments (both minor and major) that should be addressed :

1) The current findings, although experimentally distinct, bear strong relevance to a previous study (<https://www.science.org/doi/10.1126/sciadv.abq2811>),

where a TR-PECD was observed during the photodissociation of a chiral molecule, resulting in achirality. Notably, both studies exhibit a comparable feature, with significant PECD resulting from scattering events taking place beyond

4 angstroms internuclear distance. It is recommended to acknowledge and reference this related work.

2) Regarding the relatively smooth variation of the PECD observed across the HOMO band in figures 4a and 4c,

it would be beneficial to discuss whether this behavior is common or uncommon. Is it attributed to the long distance of 5 angstroms, with only a few scattering events occurring before the emission of the photoelectron? Clarifying this aspect would provide valuable insights into the nature of the observed PECD behavior.

3) In reference to Figure 3, the isodensity value of 0.07 for the MO is commonly employed and considered standard.

However, it would be highly valuable to include in the supplementary material an isodensity representation at a lower value, such as 0.01. This additional illustration would help elucidate the regions and mechanisms involved in the hybridization of the MO(s) responsible for binding the molecular complex.

4) Regarding Figure 4d, it is essential to address an issue where the color plot legend is absent :
only two plots are discussed despite the display of three plots. This discrepancy is misleading and must be rectified prior to publication. Additionally, it is pertinent to mention that these plots are the result of photon energy tuning, providing further context and clarity to the reader.

5) On page 13, line 226, the phrase "These measurements unambiguously evidence induced PECD on the HOMO orbital of the hydrogen-bonded phenol"

should be revised to "These measurements unambiguously evidence induced PECD during photoionization from the HOMO orbital of

the hydrogen-bonded phenol." This change is necessary to clarify that the observed PECD arises from the photoionization process itself,

which involves the interplay of the initial wavefunction, scattering process, and the chiral continuum of the electron.

6) The size of Figure S6 is inadequate and should be increased to enhance visibility and legibility for readers.

7) For non-expert readers, it would be beneficial to provide a more detailed legend for the insert in Figure 4.

Specifically, mention that it represents the VMI (velocity map imaging) projection of the forward hemisphere

for the phenol(R)MO_x complex. This clarification will help readers understand the nature and significance of the insert.

Reviewer #3 (Remarks to the Author):

Induced Photoelectron Circular Dichroism in an Achiral Chromophore by E. Rouquet et al.

This manuscript describes a detailed combined experimental and theoretical study of how an achiral chromophore, phenol, can acquire PECD signatures when interacting with a chiral molecule, methyl oxirane. While chirality transfer induced by direct contact, for example, non-covalent interactions, has been previously reported using techniques like vibrational circular dichroism, electronic CD, and Raman optical activity, this study presents the first demonstration of induced chirality through intermolecular interactions using photoelectron circular dichroism (PECD). Specifically, this chiroptical response is

induced within the electric dipole approximation, distinguishing it from the observations made with VCD, ECD, and ROA. The results hold significant interest for fundamental advances and also offer potential analytical applications, such as determining the chiral properties of non-aromatic chiral molecules. Overall, the paper is well-written with a clear logical flow, and I strongly support its publication in Nature Communications.

Some comments/suggestions are offered below to further improve the manuscript:

1) In the introduction section on induced chirality, the authors mentioned induced chirality caused by direct contact, i.e., inter or intramolecular interactions. It is worth noting that more recently, intense induced chiral Raman signatures of achiral solvents have been reported when a chiral solute is under resonance. The mechanism behind this phenomenon is related to the multiple light-chiral matter interaction events that occur under resonance condition (Angew. Ed. Int. Ed. 2019, 58, 16495; 2020. 59, 21895.) With this mechanism, all solvent molecules in the light path contribute to the induced chiral Raman signals. Including this new transfer mechanism would be beneficial.

2) One main question is about the transmission range of chirality. In the current study, the long-range chiral scattering potential was stated to extend to intermolecular distances up to $\sim 5 \text{ \AA}$, which corresponds to the distance between the centre of the aromatic ring of the phenol moiety and the asymmetric carbon atom of MOx. It would be helpful if the authors provided a physical description of the chiral scattering potential and its relationship with the distance to the chiral carbon.

3) I found the PECD results with Phe2:MOx fascinating, even though it was not the main focus of the current study. In particular, the statement "This result indicates that both phenol rings are sensitive to the chirality of the MOx host, even the phenol not directly bound to it." caught my attention. It is evident that the second phenol molecule would be quite far away from the carbon stereogenic center, separated by at least two intermolecular H-bond distances. This brings us back to the question raised in 2) How can we visualize the chiral scattering potential in these systems? Is the distance from the stereogenic center important at all? Perhaps the authors can offer some insights or comments regarding this matter.

We thank the reviewers for their overall very positive assessments of this manuscript and their helpful comments, which we considered carefully. For the sake of clarity, the reviewers' verbatim comments are in black and our answer in blue. We have followed the suggestions of the Reviewers, and carefully read the manuscript to correct the last typos.

(Reviewer #1 (Remarks to the Author):

The authors describe a carefully planned and prudently performed experiment in which a photoelectron is triggered from an achiral chromophore close to a chiral molecule. To do this, they first use two-photon spectroscopy and double resonance spectroscopy to determine the structure of the molecular complex before using model calculations and photoelectron spectra to find a suitable ionization window that is uniquely attributable to ionization from the achiral chromophore. They observe a PECD that changes sign with the change of enantiomer. In addition, with reference to the supplement, it is reported that the effect can also be clearly observed at a complex with two achiral molecules. The explanation given is scattering from the chiral core with the additional intriguing aspect that the electron starts far from outside the chiral portion of the complex.

The only difficulty I felt while reading was the terminology: from the clearly explained physics, for my understanding "Induced Photoelectron Circular Dichroism in an Achiral Chromophore" is misleading, and it should instead be explained as "Induced Photoelectron Circular Dichroism triggered by an Achiral Chromophore" (or something in this spirit) to clarify that the PECD effect still comes from the chiral part of the complex. I ask the authors to seriously reconsider this suggestion and make the appropriate changes in the text prior to publication (of course, publication is recommended).

We thank the reviewer for the suggestion. To avoid any misunderstanding and emphasize the role of the chiral part of the complex in the observed PECD, we have changed the title for "Induced Photoelectron Circular Dichroism onto an Achiral Chromophore".

Two further points:

The anion PECD was first observed by Weitzel et al. and should probably be cited instead or in addition to the much later work cited here (P. Krueger and K.-M. Weitzel, *Angew. Chem., Int. Ed.*, 2021, 60, 17861–1786)

Thank you for the suggestion, we have added the reference; it is Ref 11 in the present new version.

In Figure four d, I could not find the color description of the data points either in the caption or in the text.

This is a good point. We have considerably modified the caption of Figure 4. We have in particular added the colour description.

The caption is now: "Figure 4| PES and dichroic parameter at 10.4 eV and 8.5 eV for the complex of phenol with the two enantiomers of methyloxirane. (a) The photoelectron spectra (solid line) and PECD for the complex of phenol with (*S*) MOx (red circles) and (*R*) MOx (blue triangles) are recorded in coincidence with the ions at m/z 152 at a photon energy of 10.4 eV. The zones corresponding to the HOMO and HOMO-1 of the complex are highlighted in orange and pink, respectively. The insert shows raw (left part) and Abel-inverted (right part) difference (LCP–RCP) images, corresponding respectively to the 2D projection and an equatorial slice of the 3D velocity distribution filtered at the mass of the Phe:(*R*)MOx complex, obtained with the velocity map imaging (VMI) spectrometer. The direction of the light is vertical, from the bottom to the top of the image and is indicated by an arrow. (b) Same for bare phenol, recorded in coincidence with the ions at m/z 94. The zones corresponding to the HOMO and HOMO-1 of bare phenol are highlighted in green and grey, respectively. As expected for an isolated achiral chromophore, the value of the PECD is zero. (c) Same for the complex at a photon energy of 8.5 eV. Neither bare phenol nor MOx can be ionised at this energy value. (d) Mean values of the PECD weighted by the PES intensity of the two highest occupied orbitals HOMO and HOMO-1 of the Phe:(*R*) MOx complex as a function of the electron kinetic energy recorded at the photon energies of 8.5, 9.0, 9.7 and 10.4 eV. The shown values are half the difference between those obtained for the Phe:(*R*)MOx and Phe:(*S*)MOx complexes. The colour code is the same as that used in Figures 4a-c, namely: green full circles and grey empty circles for the HOMO and HOMO-1 of phenol, respectively, and orange full squares and pink empty squares for the HOMO and HOMO-1 of the complex, respectively."

Reviewer #2 (Remarks to the Author):

The manuscript by Rouquet et al. presents an experimental investigation of photoelectron electron circular dichroism (PECD) on Phe:(chiral)MOx complexes. The photoionization occurs from the two lowest molecular orbitals (MO) localized primarily on the achiral phenol moiety. Surprisingly, despite the achirality of these initial MOs, the PECD exhibits a significant 3% amplitude, depending on the enantiomer. This finding suggests that the photoelectron asymmetry arises from the long-range region of the chiral molecular potential- in the 5 angstroms range, experienced during electron scattering before its emission.

In general, I highly value the experimental findings presented by the authors, which unquestionably surpass the novelty threshold of Nat. Com. The paper demonstrates a well-structured organization and provides a compelling discussion on the prevalence of a singular conformer in the jet. The manuscript itself is easily comprehensible, presenting a clear and coherent narrative throughout. In my opinion, this work unquestionably merits publication and holds immense potential for opening up intriguing avenues in the field of chiro-optical identification of complexes.

List of specific comments (both minor and major) that should be addressed :

1) The current findings, although experimentally distinct, bear strong relevance to a previous study (<https://www.science.org/doi/10.1126/sciadv.abq2811>), where a TR-PECD was observed during the photodissociation of a chiral molecule, resulting in achirality. Notably, both studies exhibit a comparable feature, with significant PECD resulting from scattering events taking place beyond 4 angstroms internuclear distance. It is recommended to acknowledge and reference this related work.

We thank the reviewer for mentioning this article. We have added a sentence in the conclusion referring to time-resolved experiments, which are indeed very relevant in the present case, and cited this reference (Ref 51). The sentence added in the conclusion says "Time-resolved photoelectron circular dichroism experiments would be very useful to further assess the spatial range of induced chirality in a molecular complex. Note that chiral changes upon covalent-bond dissociation of a chiral molecule have been recently observed by time-resolved PECD."

2) Regarding the relatively smooth variation of the PECD observed across the HOMO band in figures 4a and 4c, it would be beneficial to discuss whether this behavior is common or uncommon. Is it attributed to the long distance of 5 angstroms, with only a few scattering events occurring before the emission of the photoelectron? Clarifying this aspect would provide valuable insights into the nature of the observed PECD behavior.

The size of the cluster being smaller than the electron escape length, which is typically several tens of Å at these kinetic energies, we cannot talk about number of scattering events, but just phase and amplitude shifts in the partial wave expansion of the outgoing electron. As largely discussed in Ref. 6, which provides all the rational in terms of scattering and phase shifts, PECD is a pure electronic process. Therefore, in the absence of continuum resonances, such as shape resonances, and within the Franck Condon approximation, the PECD should be flat over a given band (besides of course some possible KE effects across the band, see Ref.38), which is indeed close to the observed smoothness of PECD across the HOMO band.

3) In reference to Figure 3, the isodensity value of 0.07 for the MO is commonly employed and considered standard. However, it would be highly valuable to include in the supplementary material an isodensity representation at a lower value, such as 0.01. This additional illustration would help elucidate the regions and mechanisms involved in the hybridization of the MO(s) responsible for binding the molecular complex.

Thank you for the suggestion. We included in the SI the additional representation of the isodensity, using an isovalue of 0.01 as suggested by the reviewer (see Fig. S2). This plot confirms the negligible contribution of the electron density located on the chiral part to the HOMO, which is entirely localized on the achiral phenol part. We have added the following sentence in the main text: "The electron density has been plotted with an isovalue of 0.01, *i.e.* much lower than the typical 0.07 value used here, and is shown in Fig. S2. The plot shows a tiny contribution of the σ orbital of the methyloxirane

CH interacting with the phenol aromatic ring, which can be neglected.”

4) Regarding Figure 4d, it is essential to address an issue where the color plot legend is absent : only two plots are discussed despite the display of three plots. This discrepancy is misleading and must be rectified prior to publication.

We have considerably modified the caption of Figure 4. We have in particular added the colour description. We have also slightly modified Figure 4 to show the light propagation direction. (Please see response to Reviewer 1)

Additionally, it is pertinent to mention that these plots are the result of photon energy tuning, providing further context and clarity to the reader.

Thank you for mentioning this, we have modified the caption of Figure 4 accordingly (see response to Reviewer 1)

5) On page 13, line 226, the phrase "These measurements unambiguously evidence induced PECD on the HOMO orbital of the hydrogen-bonded phenol" should be revised to "These measurements unambiguously evidence induced PECD during photoionization from the HOMO orbital of the hydrogen-bonded phenol." This change is necessary to clarify that the observed PECD arises from the photoionization process itself, which involves the interplay of the initial wavefunction, scattering process, and the chiral continuum of the electron.

Thank you for the suggestion, which we have followed. We have included the suggested sentence.

6) The size of Figure S6 is inadequate and should be increased to enhance visibility and legibility for readers.

We are sorry for that; it was a question of pdf conversion. The problem has now been solved.

7) For non-expert readers, it would be beneficial to provide a more detailed legend for the insert in Figure 4. Specifically, mention that it represents the VMI (velocity map imaging) projection of the forward hemisphere for the phenol(R)MOx complex. This clarification will help readers understand the nature and significance of the insert.

Thank you for mentioning this. We have modified the caption of Figure 4 and mentioned the enantiomer used for recording the image and the direction of the light, which now also appears in Figure 4 as an arrow. Please see also the response to Reviewer 1.

Reviewer #3 (Remarks to the Author):

Induced Photoelectron Circular Dichroism in an Achiral Chromophore by E. Rouquet et al.
This manuscript describes a detailed combined experimental and theoretical study of how an achiral chromophore, phenol, can acquire PECD signatures when interacting with a chiral molecule, methyl oxirane. While chirality transfer induced by direct contact, for example, non-covalent interactions, has been previously reported using techniques like vibrational circular dichroism, electronic CD, and Raman optical activity, this study presents the first demonstration of induced chirality through intermolecular interactions using photoelectron circular dichroism (PECD). Specifically, this chiroptical response is induced within the electric dipole approximation, distinguishing it from the observations made with VCD, ECD, and ROA. The results hold significant interest for fundamental advances and also offer potential analytical applications, such as determining the chiral properties of non-aromatic chiral molecules. Overall, the paper is well-written with a clear logical flow, and I strongly support its publication in Nature Communications.

Some comments/suggestions are offered below to further improve the manuscript:

1) In the introduction section on induced chirality, the authors mentioned induced chirality caused by direct contact, i.e., inter or intramolecular interactions. It is worth noting that more recently, intense induced chiral Raman signatures of achiral solvents have been reported when a chiral solute is under resonance. The mechanism behind this phenomenon is related to the multiple light-chiral matter interaction events that occur under resonance condition (Angew. Ed. Int. Ed. 2019, 58, 16495; 2020.

59, 21895.) With this mechanism, all solvent molecules in the light path contribute to the induced chiral Raman signals. Including this new transfer mechanism would be beneficial.

Thank you for mentioning this work. We have cited this work (Ref 18) and added the following sentence in the introduction: "It has been postulated as well in the case of Raman Optical Activity in an achiral solvent".

2) One main question is about the transmission range of chirality. In the current study, the long-range chiral scattering potential was stated to extend to intermolecular distances up to $\sim 5 \text{ \AA}$, which corresponds to the distance between the centre of the aromatic ring of the phenol moiety and the asymmetric carbon atom of MOx. It would be helpful if the authors provided a physical description of the chiral scattering potential and its relationship with the distance to the chiral carbon.

A simple image consists in relating the efficiency of the scattering process to the solid angle viewed by the departing photoelectron. The efficiency therefore varies as a function of $1/r^2$, r being the distance to the chiral centre. In this context, it is expected that the PECD effect decreases when the distance r increases, and vanishes at very long distance. One way of rationalizing, in a more quantum mechanics-oriented picture, is to consider that the electron wavefunction will be chiral if the molecular potential is not equivalent in the azimuthal coordinate, so that both even and odd partial waves coexist. If this happens at long distances from the emitter, then the relative amplitude of these interferences will be negligible and the effect will disappear. To address this point, we added Ref 30 in the revised version of the manuscript, together with the following sentence: "A recent report of photoelectron elliptical dichroism of chiral aromatic alcohols showed that the dichroic effect decreased with increasing distances between the benzene moiety from which the photoelectron departs and the chiral centre." The extend of the PECD range is a very interesting point, which is precisely one of the main outcome of our article. A precise determination of the interaction range requires a detailed theoretical approach, which is beyond the scope of this work, but we are aware that several studies, performed by several groups, are in progress to investigate this aspect.

3) I found the PECD results with Phe2:MOx fascinating, even though it was not the main focus of the current study. In particular, the statement "This result indicates that both phenol rings are sensitive to the chirality of the MOx host, even the phenol not directly bound to it." caught my attention. It is evident that the second phenol molecule would be quite far away from the carbon stereogenic center, separated by at least two intermolecular H-bond distances. This brings us back to the question raised in 2) How can we visualize the chiral scattering potential in these systems? Is the distance from the stereogenic center important at all? Perhaps the authors can offer some insights or comments regarding this matter.

In fact, the second phenol molecule is not that far from the MOx molecule, hence the chiral centre. As seen in Fig. S6, the distance between each phenol molecules and the chiral centre is about the same. What is remarkable is that the second phenol is not H-bonded to MOx.

REVIEWERS' COMMENTS

Reviewer #2 (Remarks to the Author):

The few modifications made to the manuscript enhance the quality and understanding of the work. The authors have responded perfectly and clearly to the various questions from the reviewers. Without any hesitation, I recommend "Nature Communications" to publish this work, which will have a strong impact on the chiral research community.

Reviewer #3 (Remarks to the Author):

I am happy with the revision by the authors.

For point 3, I was initially confused by the statement "even the phenol not directly bound to it." and derived from that statement that the second phenol molecule was far from the stereogenic center. It was difficult to judge in Figure S6 the distance between the second phenol molecule and the stereogenic center, C*.

"What is remarkable is that the second phenol is not H-bonded to MOx." There may be some aromatic pi--HC interaction so that the second phenol molecule is also bound to the host.

We thank the reviewers for their appraisal of this manuscript. For the sake of clarity, the Reviewers' verbatim comments are in black and our answer in blue. We have slightly modified the text according to Reviewer's # 3 comments.

REVIEWERS' COMMENTS

Reviewer #2 (Remarks to the Author):

The few modifications made to the manuscript enhance the quality and understanding of the work. The authors have responded perfectly and clearly to the various questions from the reviewers. Without any hesitation, I recommend "Nature Communications" to publish this work, which will have a strong impact on the chiral research community.

We thank Reviewer #2 for these positive comments.

Reviewer #3 (Remarks to the Author):

I am happy with the revision by the authors.

For point 3, I was initially confused by the statement "even the phenol not directly bound to it." and derived from that statement that the second phenol molecule was far from the stereogenic center. It was difficult to judge in Figure S6 the distance between the second phenol molecule and the stereogenic center, C*.

"What is remarkable is that the second phenol is not H-bonded to MOx." There may be some aromatic pi--HC interaction so that the second phenol molecule is also bound to the host.

We thank Reviewer #3 for discussing this interesting point. We have slightly modified the text of the Supplementary Information. We have specified that the 2nd phenol molecule was not hydrogen bonded to the MOx molecule. We have added the distances from the centre of the aromatic ring of the 2nd phenol molecule to the stereogenic centre and to the CH of the MOx molecule, which seems, as suggested by Reviewer #3, to be involved in a CH... π interaction. The text of the Supplementary Information now says: "Despite the fact that the phenol sub-unit acting as a donor in the phenol dimer is not directly hydrogen-bound to the MOx molecule, the distance from the centre of its aromatic ring to the latter is only 4.8 Å. Moreover, a weak CH... π interaction takes place with a distance between CH and the centre of its aromatic ring of 2.9 Å. "

and later in the Supplementary Information:

"This result indicates that both phenol rings are sensitive to the chirality of the MOx host, even the phenol not directly hydrogen-bound to it but interacting through a much weaker CH... π interaction".